# Multi-omics analysis indicates an association between TAPBP and prostate cancer

Xinlong Wang[1,2☯], Aimin Jiang[2☯], Chao Li[1*], Zhiyong Liu[2*]

1 Department of Urology, The Third Xiangya Hospital of Central South University, Changsha, China,
2 Department of Urology, Changhai Hospital, Naval Medical University, Shanghai, China

☯ These authors contributed equally.
* li4307chao@126.com (CL); medliuzy@163.com (ZL)

## Abstract

Prostate cancer is one of the most common malignant tumors among men worldwide, and surgery remains its mainstay of treatment. It is unclear how prostate cancer develops and what the most effective drug targets are for treating prostate cancer. Therefore, we sought to identify the genes responsible for prostate cancer. By integrating multidimensional and high-throughput data, proteome wide association studies (PWAS), transcriptome wide association studies (TWAS), single-cell sequencing, functional enrichment, Mendelian randomization (MR), and Bayesian co-localization analyses were used to screen for candidate genes that may contribute to prostate cancer and associate with clinical results of prostate cancer. Our comprehensive analysis showed that protein abundance of eight genes was associated with prostate cancer, four of which were validated at the transcriptome level. These 8 candidate genes (MSMB, PLG, CHMP2B, ATF6B, EGF, TAPBP, GAS1 and MMP7) were validated. After combining single-cell sequencing, Mendelian randomization, and Bayesian co-localization analyses, we identified 1 gene (TAPBP) that is strongly associated with prostate cancer.

## Introduction

Prostate cancer (PCa) is one of the most common malignancies among men worldwide. According to Global Cancer Statistics 2020, PCa is the leading cause of cancer-related death in men [1]. Numerous PCa-associated risk loci have been screened using the genome-wide association study (GWAS) approach [2]. GWAS aims to tightly link genetic loci to diseases and other genetic traits [3,4]. Over the past decade, the approach has involved many variant-phenotype associations [5] and has driven important scientific discoveries [6].

Despite the tremendous impact of GWAS, inherent difficulties continue to limit its success [7,8]. An improvement of these methods would be a protein-centered approach that considers the impact of genetic variation on gene function. Proteins

**Data availability statement:** All relevant data are within the manuscript and its Supporting Information files.

**Funding:** This work was supported by the The National Natural Science Foundation of China (NSFC 82303120) from CL and Family Planning Special Project (No. 23JSZ03) from LZY. All costs associated with this manuscript, including publication fees, are covered by these funding sources. There was no additional external funding received for this study. The funders had no role in study design, data collection and analysis, decision to publish, or preparation of the manuscript.

**Competing interests:** The authors declare that they have no competing interests.

are the most effective biomarkers and therapeutic targets [9,10] because they represent the major functional components of cellular and biological processes as well as the end products of gene expression [11]. The study of risk proteins in PCa is crucial. Overall, studies have shown that alterations in proteomics are associated with cell cycle control, DNA repair, proteasome degradation and metabolic activity. In addition to differences in study methodology, possibly due to the heterogeneity of PCa itself, the results are rather inconsistent [12].

Based on the available literature, we used proteome wide association studies (PWAS). PWAS is based on the premise that causal variation in coding regions affects phenotype by altering the biochemical function of a gene's protein product. To capture these effects, PWAS quantifies the extent to which proteins are impaired under individual genotypes.

Like other gene-based approaches, PWAS alleviates the burden of multiple test corrections. In addition, it provides specific functional explanations for the protein-coding genes it discovers. Therefore, we sought to identify new drug targets for the treatment of PCa by combining high-throughput proteomics in PCa with genetic data to determine the levels of genomic structure-associated proteins. To identify potential protein biomarkers, we took a six-step approach to systematically link protein biomarkers to PCa. First, we used findings from the Atherosclerosis Risk in Communities (ARIC) dataset and prostate cancer GWAs to perform PWAS and its FDR-corrected analysis. Next, we analyzed the expression of PWAS-identified risk genes by the Prostate Cancer Single Cell Database and explore the potential function of candidate genes. The we explored important genes driving PWAS signaling at the transcriptional level by utilizing gene expression data. Fourth we used independent Mendelian randomization (MR) analysis to validate PWAS significant genes and used COLOC to integrate the GWAs data with the PCa database using Bayesian co-localization analysis to explore whether the two correlated signals were consistent with a common dependent variable(s). Fifth, we explored the mutation and drug sensitivity of TAPBP. Finally, we explored the association of these potential prostate cancer-causing proteins with some PCa clinical features.

## Method

### 1  Source of data

**1.1  Source of GWAS data.**  A large number of meta-analyses have been conducted on PCa GWAS, including the UK biobank (UKB) data are fully available at http://fastgwa.info/ukbimpbin and the GWAS Catalog. According to UKB and GWAS Catalog summary statistics, 7769 cases and 201039 controls were considered eligible. All participants for PCa included were of European descent.

**1.2  Human blood proteomic and transcriptomic data.**  Serum proteomic data were collected Atherosclerosis Risk in Communities (ARIC) study; N~9,000) [13] in a large population-based study. The ARIC study enrolled 15,792 participants from four American communities. The current study ultimately included 9,084 participants with plasma protein data. A total of 4,657 human serum proteins were considered eligible.

Local gene expression regulation is often shared across tissues [14,15] and combining eQTL data across multiple tissues can improve the power of TWAS. Thus, Genotype-Tissue Expression (GTEx) version 8 data (n = 948) was used for the analysis of whole blood eQTLs [16]. A paired RNA-seq (Illumina TruSeq; Illumina Inc) was used to obtain gene expression data, and whole genome sequencing was used to obtain genotype data. The GTEx website provides information on donor registration, consent procedures, methods for obtaining biological samples, sample attachments, and procedures for histopathological examinations [17].

**1.3  Single cell data.**  Single cell datasets of PCa were downloaded from GEO with the access number of GSE141445. The pre-processing standards for Seurat object were as follow: number of RNA count >= 1000, number of features >= 200, percent.mt <= 20, percent.rb<= 20. Double cells were removed by R package DoubletFinder with default parameters.

**1.4  Ethics statement.**  This study dose not involve direct research on humans or animals.

## 2  Statistical analysis

**2.1  Proteome-wide association studies (PWAS).**  The overall design of this study is shown in **Fig 1**. The effect of SNPs on protein abundance was calculated for proteins with significant heritability (P < 0.01) using FUSION [18]. The common prediction models used in the analysis are top1, blup, lasso, enet and bslmmv [18]. the ARIC database was selected for this analysis, so only enet and top1 were selected for the analysis. [13]. To combine the genetic effects of PCa with protein weights for PWAS of PCa, a linear sum of Z-scores + weights of independent SNPs was calculated. To reduce the likelihood of false positives, a Bonferroni-corrected P-value threshold was employed. Benjamini-Hochberg (BH) was also used to impute the P value adjusted for false discovery rate (FDR).

**2.2  Single cell analysis.**  R package harmony was applied to remove batch effect of samples from different patients. Cell type identification was determined by known marker genes and aid of R package SingleR. Enrichment score of GWAS related signatures for each cell were analyzed by function AddModuleScore from R package Seurat. More information about scRNA sequencing library preparation and sequence strategy can be found in work from study. In addition, more detailed parameters and pipelines for single cell analysis could refer to our previous works [19–21]

**2.3  Transcriptome-wide association studies (TWAS).**  The genetic influence of PCa was integrated with mRNA expression weights by using FUSION [18]. The basic procedure is as follows: Firstly, TWAS expression weights (i.e., SNP-gene correlations) are calculated from a reference expression panel [22]. To determine the out-of-sample $R^2$ of each gene prediction model, FUSION performed five-fold cross-validation [18].

**2.4  Mendelian Randomization (MR) analysis.**  The PWAS significant genes (from the FUSION method) were examined for cis-regulatory protein abundance using MR. The genome-wide significant (P < 5 × 10⁻⁶) SNPs were considered as inclusion and followed by linkage disequilibrium (LD) to screen for independent SNPs (R2 < 0.1). A harmonization of exposure (QTL) and outcome (PCa GWAS) data was then performed. Wald ratios can be used to estimate causality when only one independent QTL is available. In cases where multiple SNPs are available, the ratios of SNP exposures to SNP outcomes were estimated using the inverse variance weighting (IVW) method for random-effects meta-analysis. In addition, when the number of SNPs exceeded three, horizontal pleiotropy was tested using the MR-egger method. Bonferroni correction thresholds for the number of genes analyzed were set at P < 0.05/multiple comparisons [23,24]. Mendelian randomization analysis was conducted using "TwoSampleMR" version 0.5.5 in R version 4.0.

**2.5  Bayesian colocalization analysis.**  The Coloc Bayesian test was used to determine the probability that the PCa risk locus and the pQTL share the same causal signal [25,26]. In prostate disease, p1 represents the probability that a specific variant correlates with a significant pQTL, p2 represents the likelihood that a particular variant will be associated with one in the disease. An individual variant's p12 is the probability of it being both an outcome and a pQTL in prostate disease. Five mutually exclusive hypotheses were tested: H0, no relation with either GWAS or pQTL; H1, relation with GWAS and no relation with pQTL; H2, relation with pQTL and no relation with GWAS; H3, relation with GWAS and pQTL,

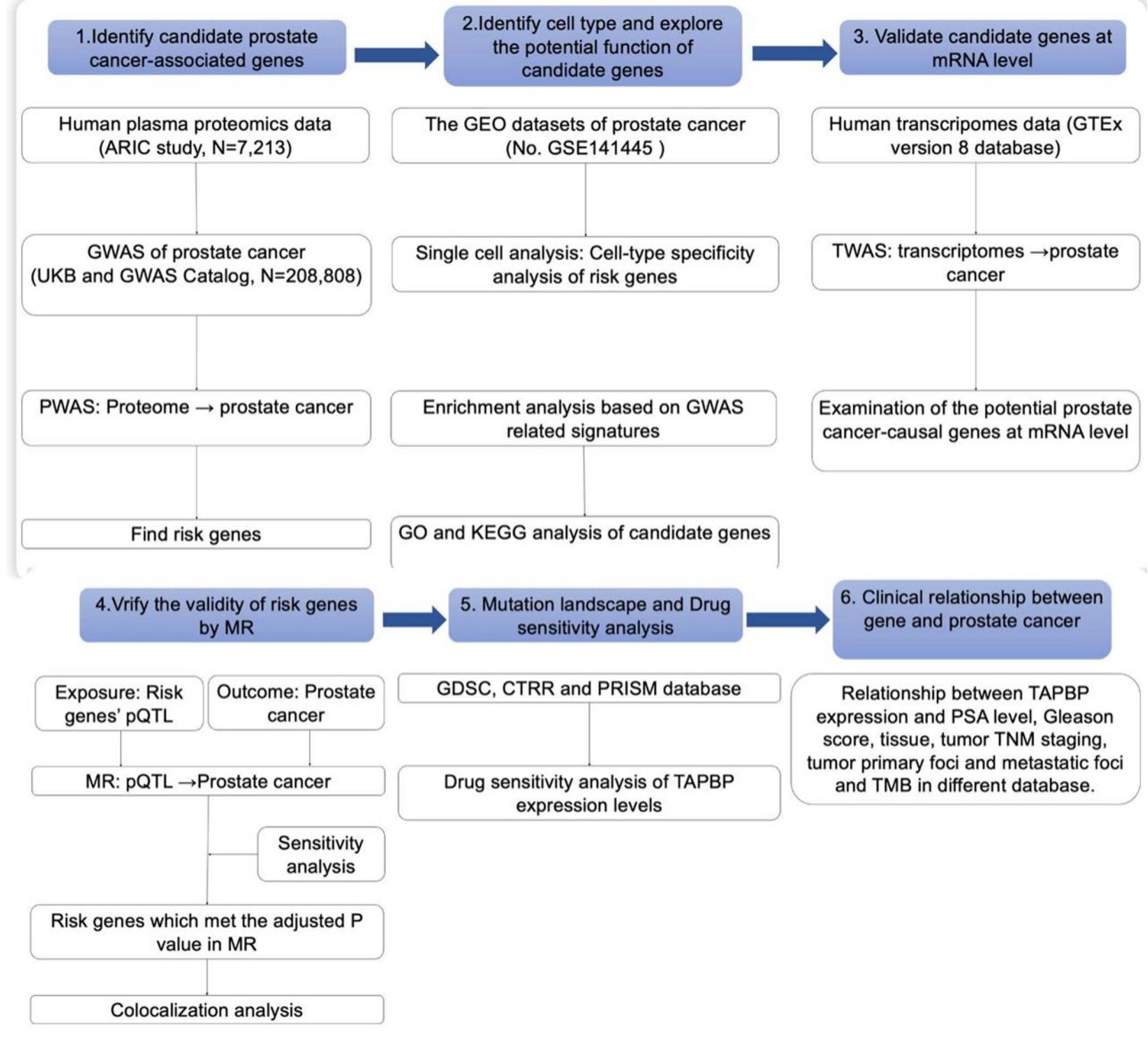

**Fig 1. Overview of this study.**

two independent SNPs; and H4, association with GWAS and pQTL, one common SNP. We focus on the last hypothesis, H4, and the posterior probability (PP) is employed to support H4 (denoted as PPH4). We define strong proof for co-localization when PPH4 ≥ 0.75.

**2.6 Enrichment analysis based on candidate genes.** Firstly, bulk sequencing datasets including ICGC-PRAD, TCGA-PRAD, GSE21034, GSE54460, GSE107299, GSE70768, GSE70769, GSE21034, DFZ2018 were downloaded from with the use of R packages TCGAbiolinks and GEOquary [27]. All transcriptome or microarray matrix were scaled and normalized before formal analysis. In addition, patients were excluded if missing clinical outcome information. Next, an enrichment score for each patient were calculated by R package GSVA with the function of ssGSEA [28]. Next, the

biological annotation and hallmarks enrichment score correlated with score of discovery PWAS genes were analyzed by R packages clusterprofile and ggpubr, which was verified by algorithm of rank-based analysis (ORA).

**2.7  Analysis of drug sensitivity analysis and mutation profile.**  Three drug sensitivity datasets, GDSC, CTRR and PRISM, were enrolled to detect the influence of GWAS score on therapy response or resistance by R package pRRophetic [29,30].

All visualization were performed with the application of software including R and GraphPad. This study examined the clinical disparities of GSWA score using student T and Wilcoxon tests. The R package ggpubr was employed to assess the different GWAS score across different T, Stage, and Grade classifications through the Kruskal-Wallis test.

## Result and discussion

### Discovery of PWAS in PCa

The FUSION pipeline was used to integrate the PCa GWAS findings with human blood proteomes for the PWAS of PCa. Based on the 9084-master analyzed sample size derived from the ARIC database, the PWAS identified 16 genes (MSMB, PLG, CHMP2B, ATF6B, EGF, TAPBP, GAS1, MMP7, SERPINA3, AIF1, PRDX3, DKK3, PSAPL1, MINPP1, ANGPTL4, and CTSS), whose protein levels were associated with PCa at P < 0.05 (**Fig 2**). After Bonferroni correction threshold of P < 0.05/number, where Microseminoprotein-beta (MSMB), Plasminogen (Plg), CHMP2B, Activating transcription factor 6 (ATF6), Epidermal Growth Factor (EGF), Tapasin Binding Protein (TAPBP), Growth Arrest Specific Gene 1 (GAS1) and matrix metalloproteinase 7 (MMP7) were more significantly associated with PCa.

### Expression of these genes in different cell types of PCa

We investigated whether risk genes identified by PWAS are enriched in specific prostate cell types. Using human single-cell RNA-seq data from a cell-type database, we identified enriched causal genes expressed specifically in eight cell types (**Fig 3**). TAPBP, ATF6B, CHMP2B, MSMB were expressed in a higher proportion of malignant cells, while TAPBP was expressed in a higher proportion and level in various immune cells.

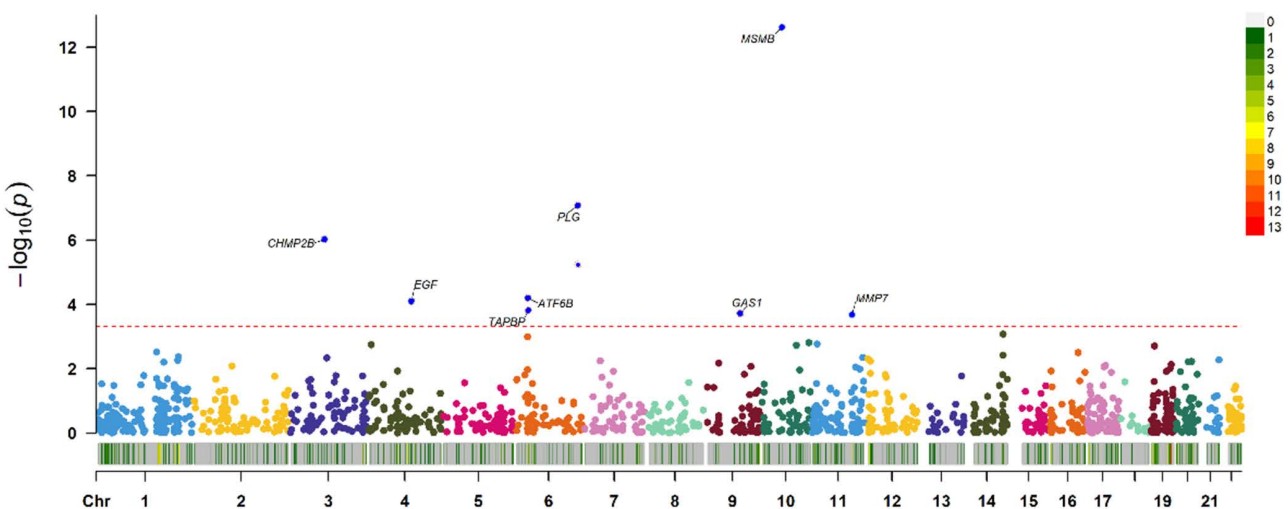

**Fig 2.  Manhattan plot for the discovery PCa PWAS integrating the PCa GWAS (N = 7769) with the discovery ARIC proteomes (N = 15792).** Each point represents a single association test between a gene and PCa ordered by genomic position on the x axis and the association strength on the y axis as the $-\log_{10}(P)$ of a z-score test. The discovery PWAS identified 8 genes whose cis-regulated protein abundance was associated with PCa at an FDR of P < 0.05. The red horizontal line reflects the significant threshold of the FDR P < 0.05 and is set at the highest unadjusted P value that is below that threshold (P = 2.2 × 10⁻⁴).

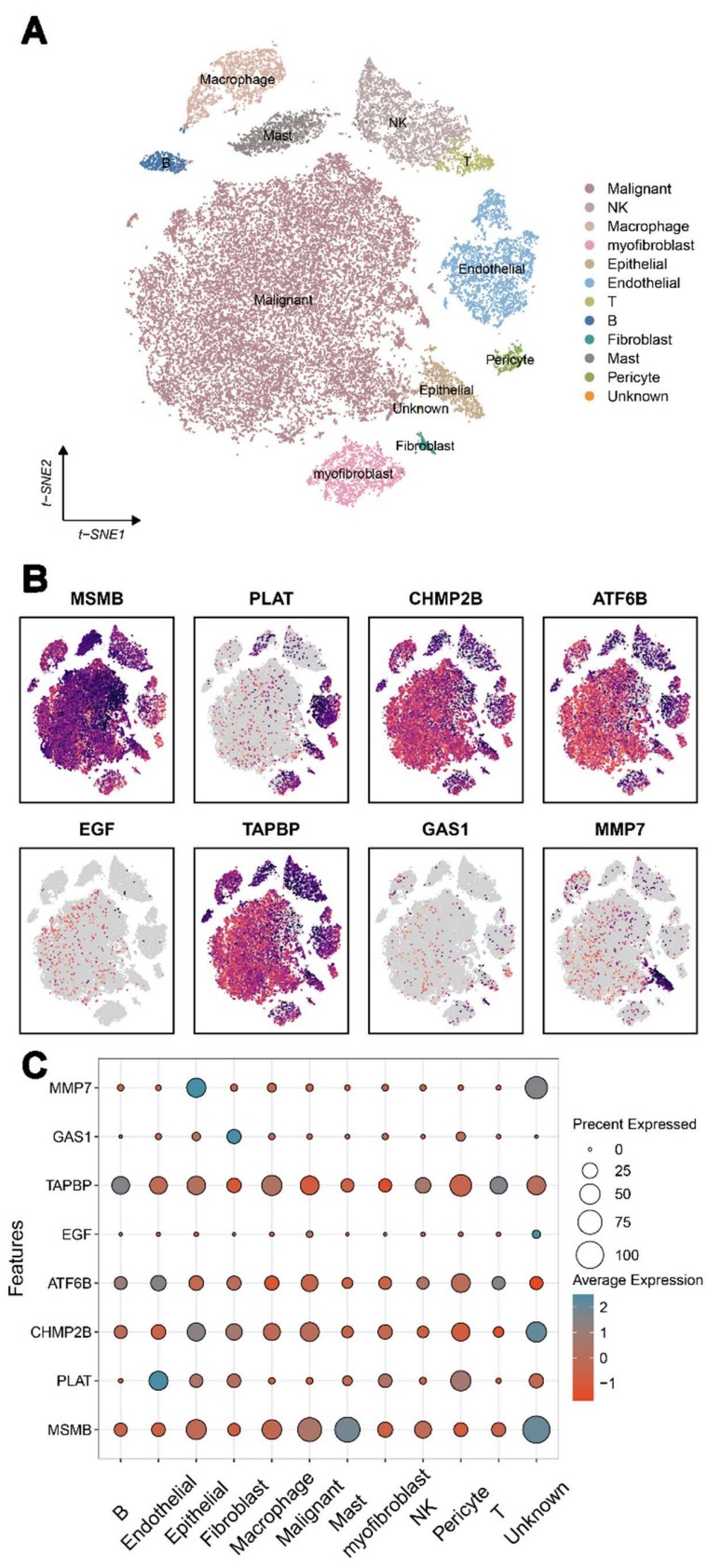

Fig 3. Expression of 8 genes in PCa tissues. A. Cell types in PCa tissues. B, C. Expression profiles of 8 genes in PCa tissues.

## Enrichment analysis of the genes in PCa

In addition, we utilized the overrepresentation analysis (ORA) algorithm to decipher the role of the PWAS discovery genes across different PCa cohorts. The GO results illustrated that PWAS discovery genes participated in cell adhesion, response to stimulus and nuclear division in BP; extracellular exosome, extracellular organelle and centromeric region in CC; and signaling receptor binding, protein−containing complex binding, and cell adhesion molecule binding in MF (Fig 4A). KEGG analysis revealed that PWAS discovery genes might be involved in cell cycle, and p53 signaling pathway (Fig 4B). GSEA and hallmark analysis also confirmed that PWAS discovery genes mainly participated in immune response, antigen processing and presentation, T cell receptor signaling pathway and Tnfa signaling via nfkb (Fig 4C, D). Through the above finding, we found that PWAS discovery genes were involved in the immune response and cell cycle. Integrating

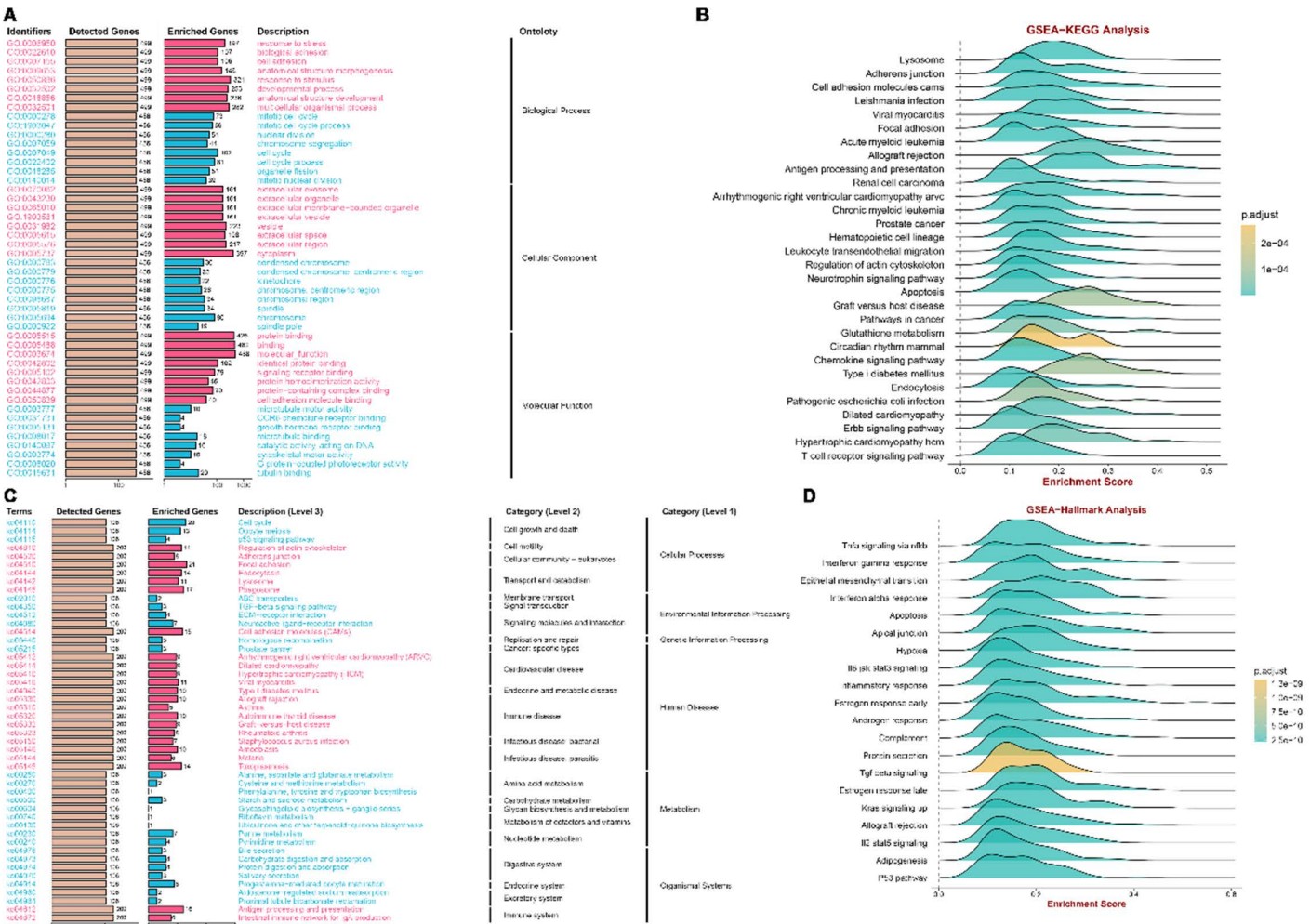

Fig 4. Candidate genes impacted signature in PCa based on correlation analysis. A, B. GO and KEGG analyses based on ORA. C, D. GSEA-KEGG, and GSEA-Hallmarks analysis of candidate genes in PCa.

the aforementioned results, we find that TAPBP is positive in all outcomes. Therefore, we will conduct further analysis on TAPBP.

## TWAS identified four genes associated with PCa

We combined PCa GWAS data with the human prostate transcriptome to perform transcriptome-wide association analysis (TWAS) of PCa using FUSION. Expression of 4 out of 8 genes (CHMP2B, ATF6B, TAPBP, GAS1) in blood was associated with PCa (Table 1). This suggests that the combined evidence from PWAS and TWAS suggests a role in the pathogenesis of PCa.

## 8 genes associated with PCa were validated by MR using prostate pQTL

Most of the analyzed proteins could only be detected using a single SNP; therefore, MR estimation was mainly based on the Wald ratio method. We further identified the above eight proteins, and these biomarkers revealed significant evidence of association in PCa GWAS (Table 2). We further analyzed their odds ratios and showed that among them, ATF6B, CHMP2B, GAS1, and MMP7 were the risk factors for the occurrence of PCa, while EGF, MSMB, PLG, TAPBP were protective factors (Fig 5).

## Co-localization between PCa risk genes and pQTL in the prostate gland

Prostate cancer PWAS associations may arise from coincidental overlap between pQTL and loci that are in linkage disequilibrium with PCa GWAS loci, or from the concurrent occurrence of a variant associated with protein expression, which is a protein quantitative trait locus (pQTL), and PCa. Statistical co-localization analyses for each gene report the probability of GWAS and pQTL sharing a causal variant, referred to as Hypothesis 4 (H4) and PP4/(PP3 + PP4) ≥ 0.75. Based on H4 ≥ 75% and PP4/(PP3 + PP4) ≥ 0.75, the analysis revealed three of the eight genes that provided evidence of genetic co-localization (MSMB, TAPBP, and EGF) (Table 3). This suggests that these three proteins play an important role in the

**Table 1.** TWAS identified four genes associated with Pca.

| gene | P value |
|------|---------|
| CHMP2B | 3.66E-04 |
| ATF6B | 1.68E-04 |
| TAPBP | 3.72E-03 |
| GAS1 | 3.85E-01 |

**Table 2.** Candidate genes identified by Mendelian randomization.

| Gene | nsnps | IVW | | | | MR Egger pval | Weighted median pval |
|------|-------|-----|----------|----------|------|---------------|----------------------|
| | | OR | ORLower | ORUpper | Pval | | |
| ATF6B | 11 | 1.2966 | 1.0964 | 1.5334 | 0.002397 | 0.630797082 | 0.000124905 |
| CHMP2B | 21 | 1.1427 | 1.0800 | 1.2091 | 3.63E-06 | 0.160124428 | 2.18935E-05 |
| EGF | 12 | 0.8494 | 0.7963 | 0.9061 | 7.21E-07 | 0.020274016 | 2.29141E-05 |
| GAS1 | 4 | 1.2076 | 1.0588 | 1.3773 | 0.004923 | 0.774952735 | 0.096094973 |
| MMP7 | 20 | 1.0962 | 1.0434 | 1.1517 | 0.000265 | 0.49333611 | 0.010928317 |
| MSMB | 7 | 0.7767 | 0.6945 | 0.8688 | 9.74E-06 | 0.327374195 | 0.000347019 |
| PLG | 23 | 0.8492 | 0.8066 | 0.8940 | 4.61E-10 | 0.017827813 | 2.70502E-07 |
| TAPBP | 24 | 0.8942 | 0.8620 | 0.9276 | 2.35E-09 | 0.004858965 | 1.9132E-07 |

pathophysiology of PCa. TAPBP showed positive results in all of the above analyses, so its relationship with PCa needs to be further explored.

## Mutation landscape between subtypes

The detailed genomic landscape difference between subgroups is depicted in **Fig 6A**, which indicates that the low TAPBP led to a high mutation frequency of speckle type BTB/POZ protein, SPOP, gain of 8q24.21 and 11q13.2 chromosome, and loss of 19q13.2 and 19q13.2 chromosome.

## Drug sensitivity analysis

We harnessed three comprehensive drug sensitivity databases, GDSC, CTRP, and PRISM, to detect the relationship between TAPBP expression and therapy sensitivity or resistance. As **Fig 6B** illustrated, high FDX1 expression displayed a consensus result of therapy resistance to Amuvatinib, Trichostatin, Pilaralisib, EHT−1864, and Panobinostat in the GDSC database; MK−2206, neuronal differentiation inducer III, lomeguatrib, BRD−A02303741:carboplatin (1:1 mol/mol), and necrostatin−7 in the CTRP database; broxaterol, albuterol, GSK1904529A, pimobendan, thiamphenicol, eprinomectin, etc., in the PRISM database.

## Potential prostate cancer-causing proteins with clinical associations

In furtherance of the understanding of the clinical association of TAPBP, a number of specific PCa clinical indicators were used to correlate studies with it. As the most commonly used PCa screening indicator, PSA was first used for analysis. We found that the expression levels of TAPBP were not significantly associated with normal PSA (**Fig 7A**). Biopsy is the gold standard for PCa diagnosis, and Gleason score is usually used to determine the benignity or malignancy of biopsied

| Exposure | SNPs | P−value | P−het | P−pleio | | OR (95% CI ) |
|---|---|---|---|---|---|---|
| A TF6B | 11 | 2.40×10−3 | 6.93×10−7 | 0.556 | | 1.30 (1.10 to 1.53) |
| CHMP2B | 21 | 3.63×10−6 | 1.27×10−2 | 0.418 | | 1.14 (1.08 to 1.21) |
| EGF | 12 | 7.21×10−7 | 9.31×10−1 | 0.618 | | 0.85 (0.80 to 0.91) |
| GAS1 | 4 | 4.92×10−3 | 2.64×10−1 | 0.903 | | 1.21 (1.06 to 1.38) |
| MMP7 | 20 | 2.65×10−4 | 1.79×10−1 | 0.241 | | 1.10 (1.04 to 1.15) |
| MSMB | 7 | 9.74×10−6 | 3.23×10−1 | 0.843 | | 0.78 (0.69 to 0.87) |
| PLG | 23 | 4.61×10−10 | 5.10×10−3 | 0.327 | | 0.85 (0.81 to 0.89) |
| T APBP | 24 | 2.35×10−9 | 5.15×10−1 | 0.720 | | 0.89 (0.86 to 0.93) |

0.6 0.8 1 1.2 1.4 1.6

**Fig 5. MR forest plot of discovered PWAS PCa-associated proteins.** The results are derived from the IVW analyses.

**Table 3. Candidate genes identified by Bayesian colocalization.**

| Gene | nsnps | PP.H3.abf | PP.H4.abf |
|---|---|---|---|
| MSMB | 596 | 0.000521504 | 0.99947791 |
| TAPBP | 5290 | 0.022416109 | 0.96049176 |
| EGF | 2841 | 0.079297104 | 0.88166609 |
| CHMP2B | 3177 | 0.99938787 | 0.00060374 |
| GAS1 | 3039 | 0.063961688 | 0.46131011 |
| MMP7 | 2986 | 0.39553352 | 0.02078762 |
| PLG | 811 | 0.994315549 | 0.00565747 |

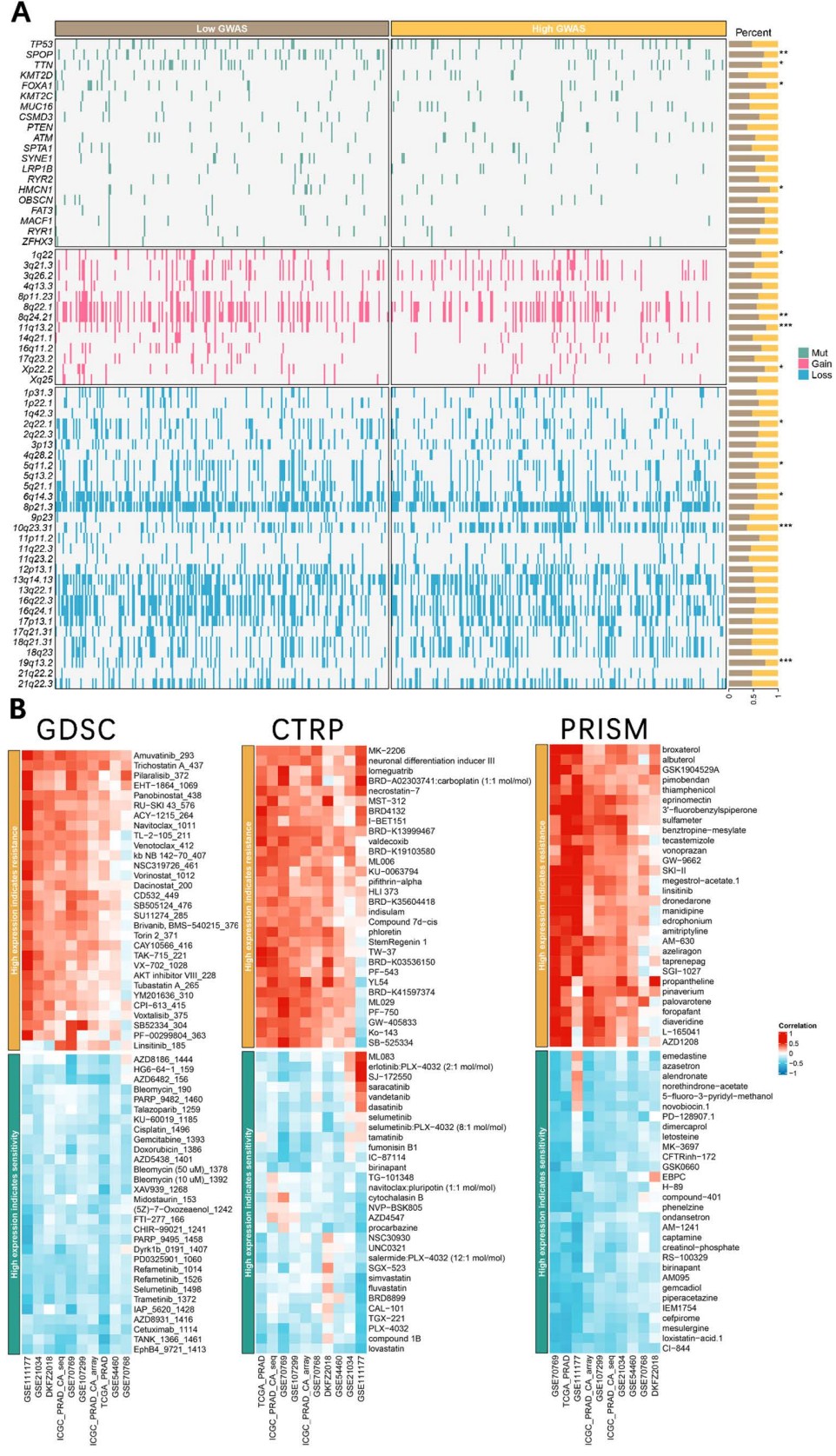

**Fig 6. A. Mutation landscape of PCa between TAPBP^high and TAPBP^low subtypes.** * p < 0.05, ** p < 0.01, *** p < 0.001, **** p < 0.0001. B. Drug sensitivity difference among different databases. Correlation of IC50 of molecular drugs and TAPBP expression levels across different databases in the GDSC database, CTRP database, and PRISM database.

tissue. Our study found that as the Gleason score increased, the expression level of TAPBP decreased (Fig 7B). Correspondingly TAPBP were more expressed overall in normal tissues than in tumor tissues (Fig 7C). Further analysis of the relationship between PCa TNM staging and the expression of TAPBP showed a negative correlation between the relative expression of TAPBP and TNM staging (Fig 7D, E, F), and showed lower expression in metastatic tumors (Fig 7G). Tumor mutational burden (TMB), as an emerging biomarker, has received increasing attention for its role in predicting the efficacy of tumor immunotherapy. High GWAS scores are associated with low TMB (Fig 7H).

## Discussion

Identifying therapeutic targets for disease is an important goal of human genetics research and is particularly important for PCA. In this study, we used a series of analytical techniques to investigate the functional associations between protein biomarkers in the prostate and PCa. We initially identified eight potential risk genes for PCa after PWAS analysis and FDR correction. Subsequently, we performed single-cell analysis of these genes and showed that they were differentially expressed in prostate. The candidate genes were validated in all of the above analyses and further enrichment analyses showed that it is primarily involved in immune response and cell cycle regulation. Further transcriptome association analysis revealed that four of the eight genes (CHMP2B, ATF6B, TAPBP, GAS1) were associated with PCa. Our results were replicated in MR validation analyses and verified whether the genes were risk factors or protective factors in the pathogenesis of PCa, providing a higher level of confidence. In addition, we identified MSMB, TAPBP and EGF by co-localization. Interestingly TAPBP was validated in all of the above analyses.

With the increasing incidence of prostate cancer in recent years [1], research on prostate cancer has also been growing. A cross-ancestry prostate cancer GWAS meta-analysis, which included 107,247 cases and 127,006 controls, reported 86 new genetic risk variants independently associated with prostate cancer risk [31,32]. A total of 269 SNPs associated with prostate cancer risk were identified in this study, including the 86 new SNPs. However, these SNPs did not include those associated with TAPBP. Nevertheless, through Mendelian randomization analysis based on GWAS data, our study found an association between TAPBP and prostate cancer. This discrepancy may be due to differences in databases and analytical methods.

TAPBP gene is close to the MHC and encodes a molecule that is a member of the IgSF. Its product, Tapasin, is required for the association of the MHC class I heterodimer with the Tap transporter protein in the endoplasmic reticulum (ER) [31]. It forms part of the peptide loading complex, which is essential for the stable assembly of class I molecules with peptides prior to transport to the cell surface.The TAPBP gene is located in the extended filamentous MHC region between the BING1 and RGL2 loci [33]. TAPBP has been recognized to play an important role in antigen presentation [34]. Our findings suggest that TAPBP is involved in the immune response and antigen presentation in prostate cancer, and it is negatively correlated with the occurrence of prostate cancer and adverse clinical features (such as higher PSA levels, higher malignancy, tumor metastasis, etc.). Based on the above results, we hypothesize that mutations or low expression of TAPBP may mediate immune evasion in prostate cancer, thereby promoting the development or progression of the disease. This hypothesis and the related mechanisms need to be further verified in subsequent experiments.

Our study has several strengths. First, PWAS for PCa was performed using the largest and most comprehensive pooled statistics from the human proteome and the most recent GWAS for PCa. Second, we performed single-cell analyses as well as enrichment analyses of the genes identified by PWAS to further validate their expression in tissues and the molecular functions that the gene products may exercise, the cellular environments in which they reside, and the biological processes in which they are involved. Third, we validated the risk proteins using independent MR validation analyses.

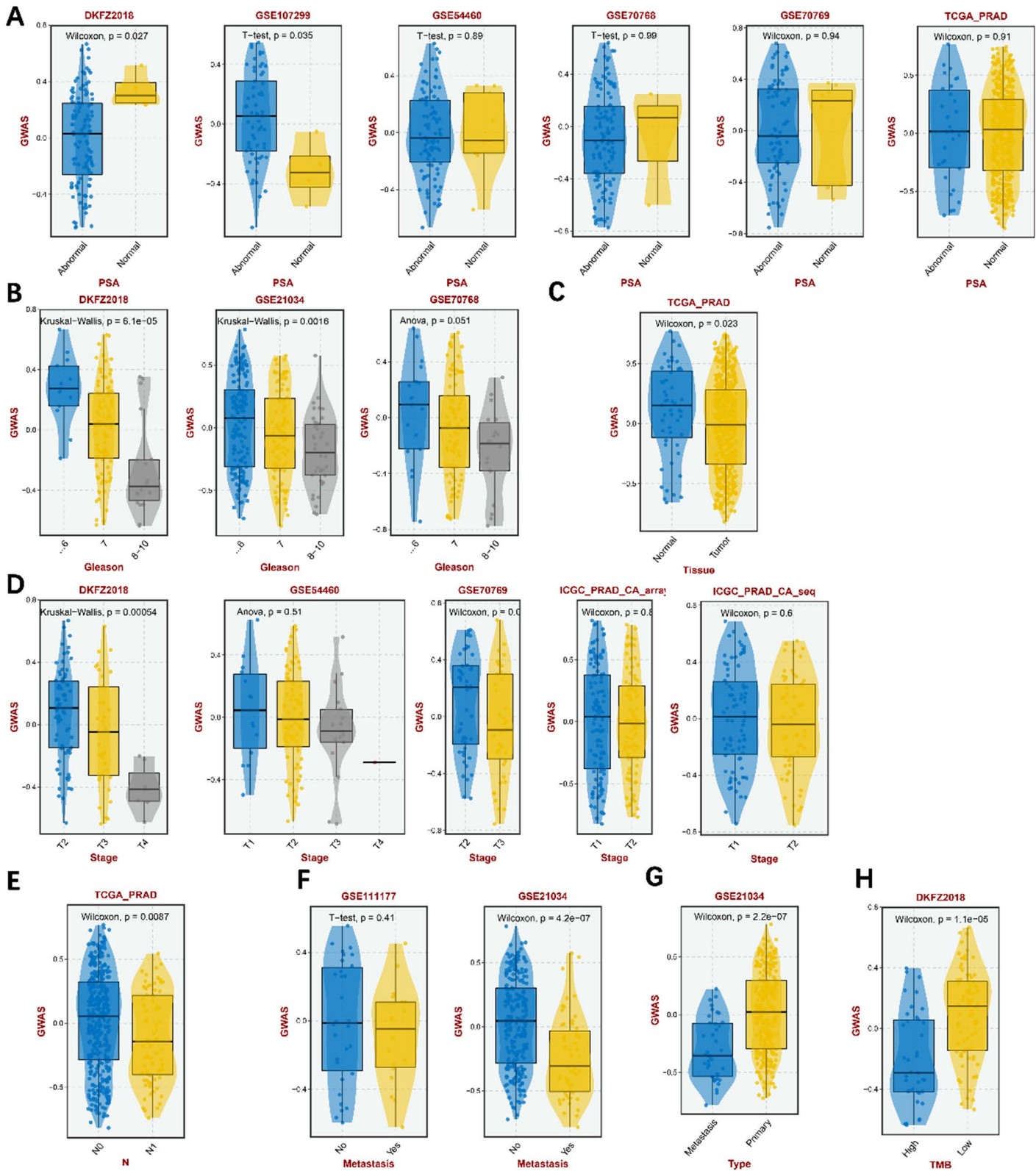

**Fig 7. The role of TAPBP in PCa patient clinical outcomes.** A. Relationship between TAPBP expression and PSA level in different databases. B. Relationship between TAPBP expression and Gleason score in different databases. C. Relationship between TAPBP expression and tissue type in

TCGA database. D, E, F. Relationship between TAPBP expression and tumor TNM staging in different databases. G. Relationship between TAPBP expression in relation to tumor primary foci and metastatic foci. H. Relationship between TAPBP expression and TMB in DKFZ2018 database.

Fourth, based on Bayesian co-localization used to estimate the probability of observing two correlated signals at specific loci with common causal variants, we confirmed the causative proteins of PCA (MSMB, TAPBP, and EGF). Finally, we explored the drug sensitivity and some clinical correlations of TAPBP, and the results indicated that low TAPBP is associated with worse clinical outcomes.

There are also some limitations to this study. First, the ARIC used in this study failed to incorporate the PSA protein. Second, including methylation data in the analysis could provide a more comprehensive picture of disease progression. Most of our studies were conducted in Europeans, but we should be careful to generalize these findings to other races. Last, the complexity of PCA biology and the molecular mechanisms behind it can only be understood through functional genomics approaches and biological experiments. Therefore, we will conduct more molecular biology experiments to validate our database-based findings.

## Conclusion

In conclusion, we have identified a number of candidate genes associated with PCa through a series of methods including PWAS, TWAS, MR, and Bayesian colocalization. They may be involved in the occurrence of PCa through immune regulation and cell cycle participation. The integrated results provide strong evidence for the association between TAPBP and PCa, and suggest that TAPBP may be a protective factor for PCa.

## Supporting information

**S1 File. MR supplement.**
(ZIP)

**S1 Checklist. STROBE-MR-checklist.**
(DOCX)

## Author contributions

**Data curation:** Xinlong Wang.

**Funding acquisition:** Chao Li.

**Project administration:** Zhiyong Liu.

**Software:** Xinlong Wang, Aimin Jiang.

**Supervision:** Zhiyong Liu.

**Validation:** Aimin Jiang.

**Visualization:** Xinlong Wang.

**Writing – original draft:** Xinlong Wang, Aimin Jiang.

**Writing – review & editing:** Xinlong Wang, Chao Li, Zhiyong Liu.

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
