## [Decision Letter · Decision Letter 0]

15 May 2025

Dear Dr. Liu,

We look forward to receiving your revised manuscript.

Kind regards,

Sarah Jose, Ph.D.

Staff Editor

PLOS ONE

Journal Requirements:

[This work was supported by the 234 Panfeng Program at Changhai Hospital (No. 2020YXK030).].

5. Please include captions for your Supporting Information files at the end of your manuscript, and update any in-text citations to match accordingly. Please see our Supporting Information guidelines for more information: http://journals.plos.org/plosone/s/supporting-information .

Reviewers' comments:

Reviewer's Responses to Questions

**Comments to the Author**

1. Is the manuscript technically sound, and do the data support the conclusions?

Reviewer #1: Yes

Reviewer #2: Partly

2. Has the statistical analysis been performed appropriately and rigorously?

Reviewer #1: Yes

Reviewer #2: Yes

3. Have the authors made all data underlying the findings in their manuscript fully available?

Reviewer #1: Yes

Reviewer #2: Yes

4. Is the manuscript presented in an intelligible fashion and written in standard English?

Reviewer #1: Yes

Reviewer #2: Yes

Reviewer #1: Relevant paper with high level of genomic-related in silico analyses. Authors showcased experience in the field of cancer genomics and modeling. However, I would recommend referring to the work of Rosalind Eeles, world expert in prostate cancer causing genes. I do not see even one of her many seminal papers cited here. So I urge the authors to amend this manuscript by referring to studies of RE, who identified prostate cancer susceptibility genes and led world-wide studies on thousands of both healthy men and individuals with prostate cancer.

Reviewer #2: The manuscript titled “Multi-omics association study identifies potential prostate cancer-causing genes” presents a multi-omics approach to identify genes potentially associated with prostate cancer, with the aim of uncovering novel biomarkers. The study is well-executed, and the results are clearly presented. However, the discussion section is notably weak and requires substantial revision. It barely engages with the study’s own findings, limiting the interpretation and contextualization of the data. Additionally, the manuscript’s title is misleading, as the results do not fully support the strong claim implied. While the study shows promise, significant improvements—particularly in the discussion—are necessary to strengthen the manuscript. Additional suggestions are provided in the annotated PDF.

**Do you want your identity to be public for this peer review?** For information about this choice, including consent withdrawal, please see our Privacy Policy

Reviewer #1: No

Reviewer #2: No

---

## [Author Response · Author response to Decision Letter 1]

12 Jun 2025

Responses to editors’comments:

Thank you for the editor’s kind suggestions. According to the editor’s requirements, we have endeavored to make the manuscript conform to the style of PLOS ONE and have included a funding statement in the new cover letter.

Responses to reviewers’comments:

Major points of Reviewer #1

Relevant paper with high level of genomic-related in silico analyses. Authors showcased experience in the field of cancer genomics and modeling. However, I would recommend referring to the work of Rosalind Eeles, world expert in prostate cancer causing genes. I do not see even one of her many seminal papers cited here. So I urge the authors to amend this manuscript by referring to studies of RE, who identified prostate cancer susceptibility genes and led world-wide studies on thousands of both healthy men and individuals with prostate cancer.

Response: We thank the reviewer for this meaningful suggestion. In the introduction section of the manuscript, we mentioned the research of Rosalind Eeles (Reference 2). Following the reviewer's suggestion, we have added a discussion of the similarities and differences with the study of Rosalind Eeles in the discussion section and provided a citation (Conti DV, Darst BF, Moss LC, et al. Trans-ancestry genome-wide association meta-analysis of prostate cancer identifies new susceptibility loci and informs genetic risk prediction. Nature genetics. 2021 Jan;53(1):65-75).

Major points of Reviewer #2

The manuscript titled “Multi-omics association study identifies potential prostate cancer-causing genes” presents a multi-omics approach to identify genes potentially associated with prostate cancer, with the aim of uncovering novel biomarkers. The study is well-executed, and the results are clearly presented. However, the discussion section is notably weak and requires substantial revision. It barely engages with the study’s own findings, limiting the interpretation and contextualization of the data. Additionally, the manuscript’s title is misleading, as the results do not fully support the strong claim implied. While the study shows promise, significant improvements—particularly in the discussion—are necessary to strengthen the manuscript. Additional suggestions are provided in the annotated PDF.

Response: We thank the reviewer for this meaningful suggestion. In the revised manuscript, we have provided a more in-depth explanation and discussion of the genes ultimately identified. Additionally, we have changed the title of the manuscript to “Multi-omics analysis indicates an association between TAPBP and prostate cancer.”

---

## [Editor Report · Decision Letter 1]

9 Sep 2025

Multi-omics analysis indicates an association between TAPBP and prostate cancer.

PLOS ONE

Dear Dr. Wang,

Thank you for submitting your manuscript to PLOS ONE. First, we apologize again for the unusual delay in processing your submission, as the original Academic Editor who had agreed to handle it became unavailable. We have invited Dr. Yang Shi, who has agreed to serve as the new Academic Editor.

**Academic Editor Comments**

**Academic Editor Comments**

We look forward to receiving your revised manuscript.

Kind regards,

Dr. Yang Shi

Academic Editor

PLOS ONE
---

## [Author Response · Author response to Decision Letter 2]

11 Sep 2025

Responses to editors’comments:

Thank you for the editor’s kind suggestions. According to the editor’s requirements, we have endeavored to make the manuscript conform to the style of PLOS ONE and have included a funding statement in the new cover letter.

Responses to editors’comments 2:

Thank you for the editor’s kind suggestions.

1. The abstract on the submission page includes non-English text; this section needs to be translated into English.

Response: We have corrected the abstract of the submission page

2. On the title page, please clearly identify the corresponding author (or state that the last two authors are joint corresponding authors, if that is what the asterisk * denotes) and provide the corresponding author’s email address.

Response: We have identified the corresponding author and email address as required

3. The title of Section 1.4, “Single-cell data,” duplicates that of Section 1.3, and the content of Section 1.4 is not re\lated to single-cell data.

Response: We have corrected the title of 1.4 to Ethics Statement

Responses to reviewers’comments:

Major points of Reviewer #1

Relevant paper with high level of genomic-related in silico analyses. Authors showcased experience in the field of cancer genomics and modeling. However, I would recommend referring to the work of Rosalind Eeles, world expert in prostate cancer causing genes. I do not see even one of her many seminal papers cited here. So I urge the authors to amend this manuscript by referring to studies of RE, who identified prostate cancer susceptibility genes and led world-wide studies on thousands of both healthy men and individuals with prostate cancer.

Response: We thank the reviewer for this meaningful suggestion. In the introduction section of the manuscript, we mentioned the research of Rosalind Eeles (Reference 2). Following the reviewer's suggestion, we have added a discussion of the similarities and differences with the study of Rosalind Eeles in the discussion section and provided a citation (Conti DV, Darst BF, Moss LC, et al. Trans-ancestry genome-wide association meta-analysis of prostate cancer identifies new susceptibility loci and informs genetic risk prediction. Nature genetics. 2021 Jan;53(1):65-75).

Major points of Reviewer #2

The manuscript titled “Multi-omics association study identifies potential prostate cancer-causing genes” presents a multi-omics approach to identify genes potentially associated with prostate cancer, with the aim of uncovering novel biomarkers. The study is well-executed, and the results are clearly presented. However, the discussion section is notably weak and requires substantial revision. It barely engages with the study’s own findings, limiting the interpretation and contextualization of the data. Additionally, the manuscript’s title is misleading, as the results do not fully support the strong claim implied. While the study shows promise, significant improvements—particularly in the discussion—are necessary to strengthen the manuscript. Additional suggestions are provided in the annotated PDF.

Response: We thank the reviewer for this meaningful suggestion. In the revised manuscript, we have provided a more in-depth explanation and discussion of the genes ultimately identified. Additionally, we have changed the title of the manuscript to “Multi-omics analysis indicates an association between TAPBP and prostate cancer.”

---

## [Decision Letter · Decision Letter 2]

26 Oct 2025

Multi-omics analysis indicates an association between TAPBP and prostate cancer.

PONE-D-24-53031R2

Dear Dr. Wang,

We’re pleased to inform you that your manuscript has been judged scientifically suitable for publication and will be formally accepted for publication once it meets all outstanding technical requirements.

Kind regards,

Yang Shi, PhD

Academic Editor

PLOS ONE

**Additional Editor Comments (optional):**

One minor issue is that some sections are numbered while others are not, which the authors are advised to correct during proofreading.

Reviewers' comments:

Reviewer's Responses to Questions

**Comments to the Author**

Reviewer #2: All comments have been addressed

Reviewer #3: All comments have been addressed

2. Is the manuscript technically sound, and do the data support the conclusions?

Reviewer #2: Yes

Reviewer #3: Yes

3. Has the statistical analysis been performed appropriately and rigorously?

Reviewer #2: Yes

Reviewer #3: Yes

4. Have the authors made all data underlying the findings in their manuscript fully available?

Reviewer #2: Yes

Reviewer #3: Yes

5. Is the manuscript presented in an intelligible fashion and written in standard English?

Reviewer #2: Yes

Reviewer #3: Yes

Reviewer #2: (No Response)

Reviewer #3: The authors have satisfactorily addressed all prior concerns. I have no additional comments. Congratulations to the authors.

**Do you want your identity to be public for this peer review?** For information about this choice, including consent withdrawal, please see our Privacy Policy

Reviewer #2: No

Reviewer #3: No

---

## [Editor Report · Acceptance letter]

PONE-D-24-53031R2

PLOS ONE

Dear Dr. Wang,

I'm pleased to inform you that your manuscript has been deemed suitable for publication in PLOS ONE. Congratulations! Your manuscript is now being handed over to our production team.

Kind regards,

on behalf of

Dr. Yang Shi

Academic Editor

PLOS ONE